# The Prevalence of and Documented Indications for Antipsychotic Prescribing in Irish Nursing Homes

**DOI:** 10.3390/pharmacy9040160

**Published:** 2021-09-30

**Authors:** Jayne E. Kelleher, Peter Weedle, Maria D. Donovan

**Affiliations:** 1Peter Weedle Allcare Pharmacy, P51HCP4 Cork, Ireland; 117223593@umail.ucc.ie (J.E.K.); pweedle@icloud.com (P.W.); 2Pharmaceutical Care Research Group, School of Pharmacy, University College Cork, T12YN60 Cork, Ireland

**Keywords:** antipsychotic medication, dementia, nursing home, prevalence, appropriateness

## Abstract

Background: Antipsychotic medications are often used ‘off-licence’ to treat neuropsychiatric symptoms and disorders of aging and to manage behavioural and psychological symptoms of dementia despite the warnings of adverse effects. Objective: To establish the prevalence of and documented indication for antipsychotic medication use in the Irish nursing home setting. Setting: This study was conducted in six nursing homes located in Co. Cork, Ireland. Method: A retrospective, cross-sectional study was employed. All patients who met the inclusion criteria (≥65 years, residing in a nursing home on a long-term basis) were eligible for inclusion. There were 120 nursing home residents recruited to the study. Main Outcome Measure: The prevalence of antipsychotic medication use in nursing home residents (with and without dementia). Results: The overall prevalence of antipsychotic prescribing was found to be 48% and patients with dementia were significantly more likely to be prescribed an antipsychotic compared to those without dementia (67% vs. 25%) (χ^2^ (1, *N* = 120) = 21.541, *p* < 0.001). In the cohort of patients with dementia, there was a trend approaching significance (*p* = 0.052) of decreasing antipsychotic use with increasing age (age 65–74 = 90%; age 75–84 = 71%; age 85 and over = 58%). An indication was documented for 84% of the antipsychotic prescriptions in this cohort. Conclusions: The findings of this study highlight that high rates of antipsychotic medication use remains an issue in Irish nursing homes. Further work should explore factors in influencing prescribing of these medications in such settings.

## 1. Introduction

As global demographic trends shift towards increased proportions of older people in populations, disorders of ageing will become a substantial challenge for those providing health and social services [1,2].

It is estimated that there are currently 33,000 people residing in nursing homes (long-term care facilities) in Ireland [3]. According to reports, approximately three-quarters of older people in Irish nursing homes are suffering from dementia [1]. It has been reported that up to 98% of people with dementia experience agitation, psychosis, or other neuropsychiatric symptoms such as anxiety, depression, and apathy at some time during the course of the disease [4,5,6]. Antipsychotic medications (APMs) are often used, mostly in an ‘off-licence’ capacity, to manage such symptoms [4,5,6]. 

Irish clinical guidelines state that where APMs are required in the treatment of behavioural and psychological symptoms of dementia, atypical APMs should be used [7,8]. Large scale meta-analyses of clinical trials have consistently demonstrated that APMs confer a 1.5 to 2 fold increased risk of mortality in people with dementia [9,10]. All typical and atypical APMs carry a black box warning from the Food and Drug Administration about this risk [9]. Atypical APMs are also linked to a 2 to 3 fold higher risk of cerebrovascular adverse events [11]. In addition, APMs are reported to cause a number of other adverse effects in people with dementia, including cardiovascular and metabolic effects, extrapyramidal symptoms, cognitive worsening, infections and falls [9,10,12]. 

Despite the warnings and the plethora of studies conducted across various settings and countries reporting adverse effects of these drugs, the use of APMs in the nursing home setting is reported to be high [13,14,15,16,17,18,19,20,21,22,23,24,25,26,27,28,29,30,31]. The prevalence of APM use in nursing homes varies significantly between countries, with reports ranging from 12% to >50% prevalence [16,18,20]. There is a general trend towards a higher prevalence of APM use in people with dementia in nursing homes compared to patients without dementia [13,20,26]. However, prevalence estimates are not sufficient to determine whether APMs are appropriately or excessively prescribed.

It has previously been stated that medicines can be considered to be prescribed ‘appropriately’ when they have a clear evidence-based indication, along with other considerations such as cost and tolerability [32]. Studies reporting the indication for APMs are limited, but suggest that off-licence prescribing of APMs is prevalent in older adults in nursing homes [32]. 

### 1.1. Aim of the Study 

To date, one study, published in 2008, determined the prevalence and appropriateness of APM use in the Irish nursing home setting. As newer prescribing guidelines and deprescribing tools have been developed since 2008, the aim of this study was to perform an updated assessment of the prevalence and potential appropriateness of APM use in nursing home residents (with and without dementia). The authors also set out to discern the indications for which APMs were most commonly prescribed in this cohort, with the ultimate goal of contributing to the development of a strategy to increase appropriateness of APM prescribing, rather than solely focusing on decreasing prevalence. 

### 1.2. Ethical Approval and Consent 

Ethical approval for this study was granted by the Clinical Research Ethics Committee University College Cork, Ireland.

Written informed consent was sought for each patient before commencing data collection. For patients who were unable to give consent, their legal representative (next of kin) were asked to consider the information on their behalf. 

## 2. Methods

### 2.1. Study Design

A retrospective, cross-sectional study design was employed. The STROBE guidelines have been followed in the conduct and reporting of this research [33]. 

For the purpose of this study, dementia was defined by at least one of the following criteria: documented use of donepezil, galantamine, memantine or rivastigmine at least once in the last two-year period, or documented diagnosis of dementia in the medical notes.

The indication for the APM was determined from the medical notes and documented. Potentially inappropriate prescribing is defined as “the practice of administering medications in a manner that poses more risk than benefit, particularly where safer alternatives exist” [34], and as stated earlier, one measure of appropriateness is determining if the APM has an evidence-based indication [32]. For the purposes of this study, atypical APMs prescribed for the treatment of persistent aggression in patients with dementia, when there is a risk of harm to self or others was considered appropriate [2,7,8]. Additionally, the prescribing of APMs to nursing home residents suffering with diagnosed psychotic disorders was also considered appropriate. The use of APM in this patient cohort outside of these criteria was documented as being potentially inappropriate. 

### 2.2. Setting

This study was conducted in six nursing homes located in Cork city and county (*n* = 449 residents). The participating homes had between fifteen and one-hundred long-term care beds, catering for long-term care, respite care and rehabilitation. 

### 2.3. Participants 

A structured convenience approach to sample selection was employed. All patients who met the inclusion criteria (≥65 years, residing on a long-term basis in any of the six homes) were eligible for inclusion. Of 435 eligible patients, 120 (28%) were recruited to the study. 

### 2.4. Data Collection, Protection and Storage

Data including patient demographics, medication lists and indication for medication, were collected between June and August 2019. This study was conducted in line with the UCC Code of Research Conduct [35]. 

### 2.5. Data Analysis 

Descriptive statistics are used to characterise the demographics and prescribing prevalence. Quantitative data were analysed using IBM SPSS (Chicago, IL, USA) version 25 and presented as frequencies or percentages. Chi-squared tests were used to determine if there was any association between categorical variables. The significance level was set as *p* < 0.05. 

## 3. Results 

### 3.1. Demographic Characteristics 

There were 120 participants in the study, which equates to a 28% recruitment rate. Ages ranged from 65 to 97 years, with the median age being 84. The majority (68%) of the study participants were female. A diagnosis of dementia was recorded in 56% of the study participants. The demographics of the study population can be found in Table 1. 

### 3.2. Prevalence

The overall prevalence of APM use in nursing home residents is 48%—this includes “as required” prescriptions as well as regular prescriptions (Figure 1). Those with dementia were significantly more likely to be prescribed an APM than those without (67% vs. 25%; χ^2^ (1120) = 21.541, *p* < 0.001). This holds true for all age groups. There was no difference in the likelihood of being on an APM, based on gender (49% females versus 46% males). In patients with dementia, a trend of decreasing APM use with increasing age was found, which approached significance in a linear-by-linear association, i.e., 90% of 65–74 years with dementia were prescribed an APM compared to 58% of dementia patients aged 85–94 years (χ^2^ (1,67) = 3776, *p* = 0.52) (Figure 2). Of the patients taking one or more APMs, 19% (*n* = 11) received an APM as required; almost half of these (*n* = 5) only received as required doses. 

### 3.3. Typical vs. Atypical APM Prescribing

A detailed breakdown of APM prescribing is provided in Table 2. Across the entire cohort, atypical APMs were prescribed more frequently than typical agents (90% vs. 10%)—this divide was mirrored for both regularly prescribed APMs and as required APMs. 

Of the atypical APMs prescribed on a regular basis, quetiapine was the most commonly prescribed APM (62%) followed by olanzapine (19%). Aripiprazole and risperidone both had prescribing rates in this cohort of 10%. Haloperidol was the most commonly prescribed typical APM (50%). 

There were a number of patients on multiple APMs on a regular basis—two patients were on two atypical APMs, one patient was on one typical and one atypical APM and one patient was on three APMs, including two atypical and one typical APM. Three of the four patients on multiple APMs had dementia.

### 3.4. Indications

A reason for the prescription of an APM was recorded in 84% of cases (Table 3). Of the documented indications, 56% of the APMs were prescribed for conditions which were not listed on the licence of the medicine; the most frequently noted off-licence indications were agitation, anxiety, depression, labile mood and insomnia. 

## 4. Discussion

The key findings from this study are that 48% of nursing home residents in six Cork nursing homes are prescribed APMs. Significantly, this study also demonstrated that patients with dementia are more likely to be prescribed one of these agents compared to those without dementia (67% vs. 25%). Comparing the findings of this study with those reported in the literature, it appears that the prescribing rates of APMs in this sample is at the higher end of the scale, with prevalences ranges from as low as 12% to as high as 64% [36,37]. This study found a trend towards a greater prevalence of APM use in younger people with dementia, in line with similar European studies [14,16,17]. 

Evidence would suggest nursing home residents are significantly more likely to be prescribed an APM than those not living in a nursing home [13,14,21]. Walsh et al. reported that patients admitted from nursing homes to acute settings in Ireland were almost five times more likely to be prescribed an APM than those who were admitted from home [34]. This culminates in a situation where those most vulnerable to adverse effects of APMs become a group particularly likely to receive them. 

Research from the Netherlands consistently report low levels of APM use [6,17,21,24,28,31,38]. Norway similarly reports low levels of APM use among nursing home residents [25]. Both of these countries have a high staff to patient ratio when caring for patients with dementia, which is a notable difference compared to other countries, including Ireland [38,39]. Qualitative research suggests that the quality of life of people with dementia increases in such environments and agitation decreases over time [39,40]. Currently in Ireland, only 1000 of the approximately 22,000 people with dementia residing in nursing homes are cared for in dementia special care units with a similar setup as Norway and the Netherlands [3]. 

Kleijer et al. investigated the variations in APM use in nursing homes in the Netherlands [21]. The authors demonstrated that variability was related to nursing home characteristics rather than differences in prevalence of neuropsychiatric symptoms among facilities. Facilities with a high prevalence of APM use were often large, situated in urban communities, and scored below average on staffing, personal care and recreational activities [21]. Further research is warranted into the influence the nursing environment has on APM use. 

Government and clinical initiatives have led to major reductions in APM use in several countries. In the UK, Banerjee concluded in 2009 that it was “time for action” in his report to the Minister of State [41]. In the space of four years (2008–2012), through the development of clinical audit tools and clinical guidelines, the UK saw a 52% reduction in the number of APMs prescribed to people with dementia [42,43]. In a recent study investigating the considerably high rates of APM use in German nursing homes, the authors claimed that prevalence remained high because reducing prescribing rates of APMs was not a priority in healthcare policy in Germany [44]. 

A number of initiatives established by governmental bodies, advocacy groups and philanthropic organisations have advanced public awareness of dementia in Ireland and led to calls for improved care for those residing in nursing homes [45,46]. The publication of the Irish National Dementia Strategy in 2014 prioritised dementia care as a matter of national policy for the first time [46]. This was considered a major milestone in how Ireland was going to respond to one of the “greatest health and social care challenges of the 21st century”. The National Clinical Guideline was published in 2019 to provide guidance to healthcare professionals on the appropriate management of psychotropic medications, including APMs in people with dementia [8]. However, our study found that the proportion of nursing home residents prescribed APMs has increased over the last decade, indicating an unfavourable upward trend [23,47,48]. 

Risperidone is the only APM currently licensed for use in people with dementia in Ireland; however, this study found that it was used less than quetiapine and olanzapine. The high rates of quetiapine prescribing in this study are striking. A large systematic review of studies reporting safety data for quetiapine in older adults, found that compared to placebo, quetiapine resulted in significantly greater cognitive impairment, higher rates of falls and injury [49]. In the CATIE-AD trials, olanzapine and risperidone were more efficacious than either quetiapine or placebo, but quetiapine and placebo were better tolerated [50]. Thus, although quetiapine may have a better safety profile than olanzapine and risperidone, recent studies have advised against prescribing of quetiapine as it has not been shown to be effective in the management of behavioural and psychological symptoms of dementia [2,51]. 

The ICGP have published a series of practice-based recommendations for appropriate APM prescribing [2,52]. The recommendations state that the vast majority of behavioural and psychological symptoms of dementia are considered to be an attempt by a person with dementia to communicate an unmet need. Therefore, a comprehensive assessment of possible precipitating factors such as pain or delirium should be performed to rule out other treatable causes, such as constipation/fear/loneliness, before considering the use of antipsychotics [2]. Some consider the use of these agents to be simply a chemical restraint, suggesting that the sedative adverse effects of APMs are used to calm people down rather than treating symptoms of behavioural and psychological symptoms of dementia [2]. This is supported by the finding that aggression was the most common indication for the prescription of APMs in this study (19%), followed by agitation (16%). 

This study examined the indication for the APM prescription in each patient. A reason for the prescription of APMs was recorded in 84% of cases. By considering off-licence/off-guideline prescribing as potentially inappropriate, just over 50% of the APMs prescribed in this study are classed as potentially inappropriate. This is not unique to Ireland; Renom-Guiteras et al. reported that potentially inappropriate prescribing was widespread in Europe [53]. 

There is scant evidence pertaining to the use of APMs as required in nursing home settings. This study reported that 19% of patients were prescribed as required APMs. Quetiapine was the most commonly prescribed as required APM (91%). Baker et al. found that 71% of clinicians had encountered occasions where as required medication had been used for reasons other than the prescribed indication [54]. The authors also claimed it was commonly found that the reason why as required APMs were administered was not recorded in case-notes, which mirrors the findings of this study [54]. 

The substantial variation in APM prescribing prevalence and indications for APM use across settings and countries highlights the challenges and opportunities for system-wide improvement. Therefore, it is imperative to examine and understand the reasons for inappropriate prescribing of APMs in order to inform the design of suitable interventions. 

## 5. Limitations

This study has several notable limitations. This study was carried out in only one county in Ireland, therefore the findings may not be generalisable to other jurisdictions. A structured convenience approach to sample selection was employed in this study. Nursing staff were responsible for handing out consent forms and therefore the possibility of selection bias cannot be excluded. Finally, our study sample was small compared to similar European studies. A larger allows for correction of independent variables that have a possible influence on APM prescribing. Despite these limitations, the present study provides a reliable and meaningful picture of the prescribing patterns of APM in the nursing home setting in Ireland.

## 6. Conclusions

This study identified that 48% of nursing home residents in six Cork nursing homes, are prescribed APMs, and that patients with dementia are more likely to be prescribed one of these agents compared to those without dementia (67% vs. 25%). This study reported potentially inappropriate prescribing of APMs among 56% of nursing home residents. 

The findings of this study highlight that high rates of APM use remains an issue in Irish nursing homes. The results clearly indicate the need for an organized and patient-centred approach towards the effective and appropriate prescribing of antipsychotic medications. It is hoped that through the implementation of the national clinical guideline, as well as interdisciplinary deprescribing processes, these goals can be achieved. 

## Figures and Tables

**Figure 1 pharmacy-09-00160-f001:**
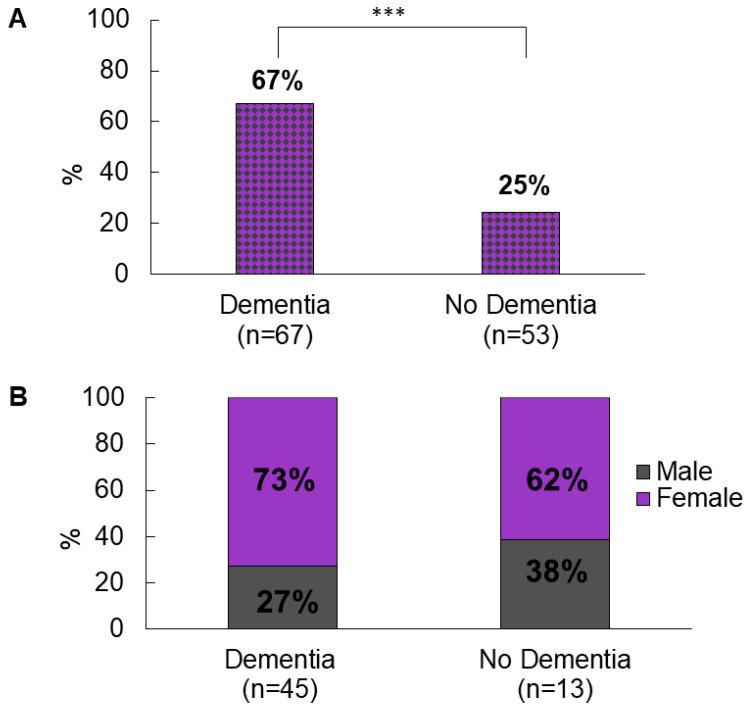
(**A**) prevalence of APM prescribing in people with and without dementia, *** *p* < 0.001. (**B**) breakdown by gender of APM prescribing in people with and without dementia.

**Figure 2 pharmacy-09-00160-f002:**
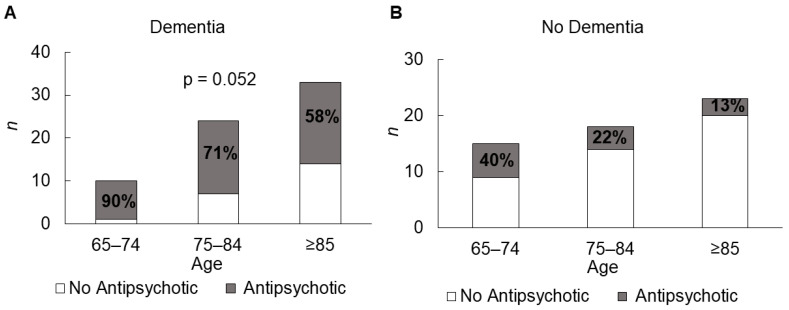
Trend of decreasing APM use with increasing age in (**A**) people with dementia and (**B**) people without dementia.

**Table 1 pharmacy-09-00160-t001:** Demographics of study population.

	Dementia (*n* = 67), *n*	No Dementia (*n* = 53), *n*	Total (*n* = 120), *n*
Gender			
Male	2	18	39
Female	46	35	81
Age			
65–74	10	15	25
75–84	24	18	42
≥85	33	20	53

**Table 2 pharmacy-09-00160-t002:** Typical vs. Atypical APM Prescribing.

ATC Code	APM	Frequency		
		Dementia (*n*)	No Dementia (*n*)	Total (*n*)
Typical (regular use)				
N05AD01	Haloperidol	343	0	3
N05AB02	Fluphenazine	0	1	1
N05AA01	Chlorpromazine	1	0	1
N05AB04	Prochlorperazine	1	0	1
Atypical (regular use)				
N05AH04	Quetiapine	24	8	32
N05AH03	Olanzapine	8	2	10
N05AX08	Risperidone	4	1	5
N05AX12	Aripiprazole	4	1	5
Typical (as required use)				
N05AB04	Prochlorperazine	1	0	1
Atypical (as required use)				
N05AH04	Quetiapine	7	3	10

Abbreviations: ATC, Anatomical Therapeutic Chemical Classification; APM, Antipsychotic medication.

**Table 3 pharmacy-09-00160-t003:** Documented Indications for Antipsychotic Prescriptions.

Indications	Total (*n*) (*n* = 69)	Dementia (*n*) (*n* = 53)	No Dementia (*n*) (*n* = 16)
Treatment of aggression	13	9	4
Treatment of agitation	11	9	2
No indication documented	11	7	4
Treatment of anxiety/depression/mood	8	4	4
Treatment of insomnia	7	7	
Treatment of schizophrenia	6	5	1
Treatment of behavioural disorders/BPSD	5	5	
Treatment of psychosis	3	3	
Delusions, hallucinations, paranoia	2	2	
Treatment of nausea/vomiting	1	1	
Treatment of sun-downing	1		1
Treatment during periods of personal care	1	1	

Abbreviations: BPSD, behavioural and psychological symptoms of dementia.

## Data Availability

The data that support the findings are available from authors, upon reasonable request.

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
