# Peer review of "The Prevalence of and Documented Indications for Antipsychotic Prescribing in Irish Nursing Homes"

_pharmacy, 2021, doi:10.3390/pharmacy9040160_

Round 1

Reviewer 1 Report

The present manuscript gives evidence of the problems of antipsychotic prescribing in older people with dementia.

Even thought the topic addressed is not a novelty, it gives data referring to a particular area of Ireland.

The work is well structured and it is easy to follow.

Some things to point out:

  • Discussion: Even though I find it very complete and informative, it is a little to long. I suggest to reduce it a little, focussing in some aspects that are somewhat repeated
  • Conclusions: When the authors point out the improvements needed, I think that it is also very important to consider the patient approach and the individualization of therapy.

Author Response

Thank you firstly for taking the time to review this work. 

The present manuscript gives evidence of the problems of antipsychotic prescribing in older people with dementia.

Even thought the topic addressed is not a novelty, it gives data referring to a particular area of Ireland.

The work is well structured and it is easy to follow.

Some things to point out:

  • Discussion: Even though I find it very complete and informative, it is a little to long. I suggest to reduce it a little, focussing in some aspects that are somewhat repeated

Thank you for your comment – we have reduced the length of the discussion, removing some sections which were not directly relevant to the research undertaken e.g. the discussion regarding non-pharmacological therapies. We have also removed repetition, where we have seen it. We hope the discussion now flows better.

  • Conclusions: When the authors point out the improvements needed, I think that it is also very important to consider the patient approach and the individualization of therapy.

We thank the review for raising this important point. We agree that adopting a patient-centred approach to care is absolutely essential in the nursing home setting. We have now added this to the conclusion:

“The findings of this study highlight that high rates of APM use remains an issue in Irish nursing homes. The results clearly indicate the need for an organized and patient-centred approach towards the effective and appropriate prescribing of antipsychotic medications. It is hoped that through the implementation of the national clinical guideline, as well as interdisciplinary deprescribing processes, these goals can be achieved.”

Reviewer 2 Report

This is a cross sectional study that tries to establish the prevalence of and documented indication for antipsychotic medication use in the Irish nursing home setting.

It is a well-structured manuscript but with few analytical results. The authors focus on writing their descriptive results without performing an analysis on the prevalence of adverse effects in all cases and perhaps an analysis according to the appropriateness or not of the prescription.

I feel that the authors make an abuse of acronyms that make it difficult to read the whole manuscript, please, it should be reviewed which ones are indispensable to leave and which ones are more appropriate to facilitate reading.

The implications of practices should be changed. They are not derived from the results of the study conducted.

I feel that the authors should revise the introduction, they should better justify the need for the study, for example by analyzing the results of the limited studies of line 74.

Could the authors add a sample size calculation? Could the sample be adequate and not convenience?

Discussion,

The Discussion should also be rephrased. I know that the authors do not have many results to discuss but they should revise the content and the actual discussion of the results:

For example:

Line 175-177: this justification is in line with what the rest of the European studies discuss?

The authors also talk about non-pharmacological therapies in the discussion. Why have they not been analyzed in the sample?

Similarly, paragraph line 234. The comorbidity could have been analyzed in the sample and thus analyze this discussion of the authors.

Could the authors add the limitations of their study? I feel there are some of them

Conclusions,

Among 50%? Where is this data previously?

All the best in your submission!

Author Response

This is a cross sectional study that tries to establish the prevalence of and documented indication for antipsychotic medication use in the Irish nursing home setting.

It is a well-structured manuscript but with few analytical results. The authors focus on writing their descriptive results without performing an analysis on the prevalence of adverse effects in all cases and perhaps an analysis according to the appropriateness or not of the prescription.

I feel that the authors make an abuse of acronyms that make it difficult to read the whole manuscript, please, it should be reviewed which ones are indispensable to leave and which ones are more appropriate to facilitate reading.

Thank you to the reviewer for highlighting to us the difficulty with reading the article with all of the acronyms. We have now removed acronyms: PwD, NPS, BPSD, prn and left just the indispensable ones. We hope the article is now easier to follow.

The implications of practices should be changed. They are not derived from the results of the study conducted.

Thank you for your comment. We have removed impacts on practice altogether, as we feel that the article flows better without them and we agree that they are not derived from the results of the study we conducted. The impact on practice of this research is now confined solely to the discussion.

I feel that the authors should revise the introduction, they should better justify the need for the study, for example by analyzing the results of the limited studies of line 74.

The authors thank the reviewer for highlighting the need to justify this study. We have now elaborated on the background to this study, to justify what this study adds to the current literature.

“To date, one study, published in 2008, determined the prevalence and appropriateness of APM use in the Irish nursing home setting. As newer prescribing guidelines and deprescribing tools have been developed since 2008, the aim of this study was to perform an updated assessment of the prevalence and potential appropriateness of APM use in nursing home residents (with and without dementia). The authors also set out to discern the indications for which APMs were most commonly prescribed in this cohort, with the ultimate goal of contributing to the development of a strategy to increase appropriateness of APM prescribing, rather than solely focusing on decreasing prevalence.” 

Could the authors add a sample size calculation? Could the sample be adequate and not convenience?

Unfortunately this is not possible. We aimed to determine the prevalence of a) APM prescribing in nursing homes in Ireland and b) how many of those APM prescriptions were potentially inappropriate. We were not carrying out an intervention in which the effect size was known and therefore, this can be considered hypothesis generating for this particular jurisdiction. The study was designed with a structured convenience sampling method and cannot be retrospectively changed. Now that we have carried out this study, we could perform a sample size calculation for future studies, based on 48% prevalence of APM prescribing and 56% potentially inappropriate APM prescriptions, depending on the aim of the study.

We have added the sample selection process as a limitation of the study design:

“A structured convenience approach to sample selection was employed in this study. Nursing staff were responsible for handing out consent forms and therefore the possibility of selection bias cannot be excluded.”

Discussion,

The Discussion should also be rephrased. I know that the authors do not have many results to discuss but they should revise the content and the actual discussion of the results:

For example:

Line 175-177: this justification is in line with what the rest of the European studies discuss?

We thank the reviewer for drawing our attention to this point. Our study, similar to three other studies, found a trend towards greater APM prescription in younger people with dementia compared to older people with dementia. However the sentence that followed is speculation and while it might be interesting to hypothesise a reason for the decrease in APM prescribing with increasing age, we do not have the evidence from our study to support it. We have now removed this sentence.

The authors also talk about non-pharmacological therapies in the discussion. Why have they not been analyzed in the sample?

The authors thank the reviewer for highlighting the discrepancy between what was studied and what was included in the discussion. As non-pharmacological therapies were beyond the scope of this study, we have now removed this paragraph in the discussion.

Similarly, paragraph line 234. The comorbidity could have been analyzed in the sample and thus analyze this discussion of the authors.

As we have now removed the paragraph about non-pharmacological treatments, line 234 from the original submission is no longer present. We agree with the reviewer that this should be removed, as it was not analysed in our study.

Could the authors add the limitations of their study? I feel there are some of them

The authors are grateful to the reviewer for pointing out this omission. There are indeed limitations to the study which have now been added:

“This study has several notable limitations. This study was carried out in only one county in Ireland, therefore the findings may not be generalisable to other jurisdictions. A structured convenience approach to sample selection was employed in this study. Nursing staff were responsible for handing out consent forms and therefore the possibility of selection bias cannot be excluded. Finally, our study sample was small compared to similar European studies. A larger allows for correction of independent variables that have a possible influence on APM prescribing. Despite these limitations, the present study provides a reliable and meaningful picture of the prescribing patterns of APM in the nursing home setting in Ireland.”

Conclusions,

Among 50%? Where is this data previously?

 Lines 165-168 of the original submission stated that:

Of the documented indications, 56% of the APMs were prescribed for conditions which were not listed on the licence of the medicine; the most frequently noted off-licence indications were agitation, anxiety, depression, labile mood and insomnia.

The authors thank the reviewer for pointing out the discrepancy between results and conclusions (56% vs 50%) and have now amended the conclusion to state 56%.

Round 2

Reviewer 2 Report

The authors have improved the manuscript